# Overground Robotic Gait Trainer mTPAD Improves Gait Symmetry and Weight Bearing in Stroke Survivors

**DOI:** 10.3390/bioengineering10060698

**Published:** 2023-06-08

**Authors:** Danielle Marie Stramel, Lauren Winterbottom, Joel Stein, Sunil K. Agrawal

**Affiliations:** 1Department of Mechanical Engineering, Columbia University, New York, NY 10027, USA; dms2281@columbia.edu; 2Department of Rehabilitation and Regenerative Medicine, Columbia University Irving Medical Center, New York, NY 10032, USA; lbw2136@cumc.columbia.edu (L.W.); js1165@cumc.columbia.edu (J.S.)

**Keywords:** robotics, gait training, stroke, force control, pelvic forces

## Abstract

Stroke is a leading cause of disability, impairing the ability to generate propulsive forces and causing significant lateral gait asymmetry. We aim to improve stroke survivors’ gaits by promoting weight-bearing during affected limb stance. External forces can encourage this; e.g., vertical forces can augment the gravitational force requiring higher ground reaction forces, or lateral forces can shift the center of mass over the stance foot, altering the lateral placement of the center of pressure. With our novel design of a mobile Tethered Pelvic Assist Device (mTPAD) paired with the DeepSole system to predict the user’s gait cycle percentage, we demonstrate how to apply three-dimensional forces on the pelvis without lower limb constraints. This work is the first result in the literature that shows that with an applied lateral force during affected limb stance, the center of pressure trajectory’s lateral symmetry is significantly closer to a 0% symmetry (5.5%) than without external force applied (−9.8%,p<0.05). Furthermore, the affected limb’s maximum relative pressure (p) significantly increases from 233.7p to 234.1p (p<0.05) with an applied downward force, increasing affected limb loading. This work highlights how the mTPAD increases weight-bearing and propulsive forces during gait, which is a crucial goal for stroke survivors.

## 1. Introduction

Stroke is the second leading cause of disability and death in adults, resulting from damage to brain cells due to a temporary blockage of blood flow to the brain or bleeding in surrounding brain tissue [1]. In 2003, 26% of new stroke survivors remained disabled in activities of daily living, and 50% had reduced mobility caused by hemiparesis [2].

Multiple deficits at the level of the skeletal musculature have been reported post-stroke, including muscle weakness, increased muscle tone and spasticity, reduced muscle coordination, and increased co-activation of antagonist muscles [3]. In addition, muscle weakness, or the inability to generate expected levels of muscle force, is associated with a decrease in muscle mass, a reduction in the muscle fiber length, and a smaller pennation angle [4]. The timing of lower limb muscle firing is also affected post-stroke, with stroke survivors showing overall increased activation and co-activity of the thigh and shank muscles for both the paretic and nonparetic lower limbs [5].

These alterations to the internal forces of the skeletal musculature post-stroke also affect the loading of each leg and the force interaction between the feet and the ground during gait. The ground reaction force (GRF) is the resultant vector combining all forces acting between the foot and the ground [6]. Stroke survivors have smaller GRFs on their hemiparetic side than on their contralateral side [7]. The severity of hemiparesis also impacts the paretic limb’s anterior–posterior GRF component and is correlated to the limb’s ability to propel the individual forward during gait [8]. Propulsive force, or the peak anterior GRF, is related to lateral weight transfer ability [9], and higher hemiparetic severity is correlated to lower paretic limb propulsion [10]. The non-affected limb GRF is also affected post-stroke and is less than the dominant foot of age-matched neurotypical adults [7]. This altered loading is thought to cause compensatory gait postures [11].

The application point of the GRF is called the center of pressure (COP) and can be an essential tool for evaluating gait pathologies [12]. During overground gait, the COP follows a continuous trajectory, shifting between the left and right feet as weight is transferred laterally and the body is propelled forward. For stroke survivors, the COP trajectory during the affected side’s single stance (SS) is shorter than the contralateral side’s single stance COP trajectory. This shortened COP trajectory on the hemiplegic side is correlated to an individual’s walking efficiency and independence [13,14,15,16]. Other studies have looked at the trajectory of stroke survivors’ COP and have seen increased COP variability in the frontal plane while walking [17] and during quiet stance [18].

Therefore, the loading of the paretic limb and the lateral symmetry of the SS COP trajectories are essential aspects of the overground gait of stroke survivors that are correlated with walking ability and independence. The ability to rehabilitate the paretic hemibody may recover or improve coordination and mobility deficits. Gait therapy is an integral part of rehabilitation for these individuals, and methods range from functional electrical stimulation (FES) to pharmacological interventions to traditional or robotic gait training [19]. Treadmill-based gait trainings are commonly studied and can be implemented with bodyweight loading to increase static paretic GRF [20] or with auditory feedback to increase the center of loading distance of the paretic limb [21]. However, treadmill training has not shown advantages over overground gait training [22] and may not have translational benefits to the task of overground walking [23]. Including robotic gait training in the care plan can increase independence [24,25], improve gait velocity and Berg Balance Scale scores [26,27], induce clinically significant improvements in locomotor tasks such as the 10-m walk test (10MWT) and Timed-Up-and-Go (TUG) [28], and increase gait speed when using end-effector-based devices [29]. However, robotic gait therapy is not definitively better than conventional gait therapy for stroke survivors [30].

The literature on post-stroke robotic gait training is mixed [31,32,33,34,35]; some studies have shown improvements in clinical evaluations or gait kinematics, while others have not found a benefit compared to traditional gait therapy. While more work is needed to extract a definitive answer, some robotic gait trainers have shown improvements in stroke survivors’ GRF and gait characteristics [36]. In one case study, active training with the treadmill-based Lokomat substantially increased the symmetry of the vertical GRF between the paretic and nonparetic limb [37]. After a multi-session gait training trial with the treadmill-based Lopes II, participants’ paretic propulsion impulse and overall propulsion symmetry improved from their baselines [38]. A multi-session overground training with a robotic ankle-f-oot orthosis increased paretic side vertical GRFs in a group of stroke survivors [39]. Our group’s TPAD, a treadmill-based cable-driven system, improved the vertical loading force symmetry between the affected and non-affected limbs when a force in the horizontal plane was applied at the pelvis during a case study [40]. After a multi-session training with the TPAD, stroke survivors showed significantly improved stance time symmetry [41]. These studies are all treadmill based because the robotic platforms are large structures that are not portable or easily maneuverable. These size and mobility constraints restrict the robotic platforms to treadmills, which may limit improvements to overground gait. While overground robotic gait training may have increased improvement potential, it has not been shown to be better than traditional gait therapy due to the small number of trials and participants [42]. However, more work is needed to determine if overground robotic gait trainers can impact stroke survivors’ functional and kinetic measures [42,43].

While there are few existing studies investigating the effects of robotic gait training on the GRFs of stroke survivors, the impact of specific applied external forces on different characteristics of the GRF is not well understood. Furthermore, previous robotic interventions do not tailor forces to the individual’s gait and needs. Therefore, a gap exists in the current literature on an overground robotic gait trainer that can alter the user’s GRFs while tailoring the applied forces to the user’s gait in real time. The mobile Tethered Pelvic Assist Device (mTPAD), which can apply three-dimensional external forces to the pelvis [44], could change the GRF and COP in the frontal plane by providing different forces.

The mTPAD applies pelvic external forces synchronized to the user’s gait in this work [45], specifically during hemiparetic stance. Using these pelvic forces while the stroke survivor is in their hemiparetic stance, we can study the effects of these forces on the affected side. In addition, we highlight the mTPAD’s unique ability to alter specific aspects of the GRF, e.g., the underfoot pressure and the COP trajectory. This work’s objective is to showcase the mTPAD’s efficacy as an overground gait-training tool that can customize its force application to the user’s gait. By illustrating that the mTPAD can alter the overground gait of stroke survivors in a single session, this work emphasizes mTPAD’s potential as an overground gait training device that customizes force applications to the user’s specific gait and gait needs.

## 2. mTPAD Asymmetric Controller

### 2.1. mTPAD

This work uses the mTPAD to explore the effects of different forces applied to the pelvis during overground walking. The mTPAD is a cable-driven, parallel robotic platform with seven degrees of freedom (DOFs), as shown in Figure 1. To allow portability, the mTPAD was built upon a NIMBO posterior rollator. Seven custom motors housings mount Dynamixel servo motors to the frame of the posterior rollator. These subassemblies route cables from the servo motor’s cable spool to a pelvic belt worn by the user. Each cable can apply nearly 70 N per cable. Cables route from each motor to a pelvic belt worn by the user. The force and moment profile at the pelvic center can be controlled by regulating the tensions in the seven cables. The open-loop control scheme shown in Figure 1 is used.

The pelvic position and the goal three-dimensional forces and moments, or wrench, are input to a quadratic programming optimization that minimizes the cable tensions while ensuring the target wrench is used. The seven DOFs of the mTPAD provide total control of the six DOFs of the pelvic output wrench, so the mTPAD has the freedom to apply customizable forces and moments at the pelvis while the user walks overground. The quadratic programming scheme uses the following tension constraints.
(1)$$\min\;f\left(T\right):f\left(T\right)=T^{⊤}T$$
(2)$$\;JT_{ineq}=F_{ineq}$$
where *T* is the vector of cable tensions (7×1). *J* is the system Jacobian (6×7), $T_{ineq}$ is the optimized tension solution (7×1), $F_{ineq}$ is the force–moment profile associated with the optimized tension solution (6×1), $T_{min}=1$ N to ensure the cables remain taut, and $T_{max}=50$ N to ensure participant safety. Boundaries for $F_{ineq}$ are set depending on the goal wrench output to the user with the boundaries for each condition in this work shown in Figure 2. To synchronize the output wrench with the user’s gait, our lab’s DeepSole system was used to predict the gait cycle percentage of stroke survivors in real time. For this work, we will consider lateral and downward forces. Further inequality constraints are added per case as illustrated in Figure 2.

### 2.2. DeepSole System

The DeepSole system, shown in Figure 3, comprises a pair of instrumented shoes and a deep learning framework [46]. Each shoe has an instrumented insole and an electronics module with an inertial measurement unit (IMU). The instrumented insoles have three pressure sensors made with piezo-resistive e-textiles located under the heel, toe, and ball of the foot. Each shoe’s linear accelerations and Euler angles are measured within each IMU’s local coordinate frame. The sensor readings are streamed through WiFi using User Datagram Protocol (UDP) data packets.

The DeepSole’s deep learning framework maps each shoe’s raw data to the shoe’s gait cycle percentage using an Encoder–Decoder Recurrent Neural Network (RNN) model. The Encoder–Decorder RNN model (ERM) uses an Encoder module, an RNN module, a dense module, a Decoder module, and a fully connected layer with a single neuron. For each shoe, the ERM’s input was the nine raw data channels, and the output was a single value from 0 to 1, representing the gait cycle percentage prediction. Time-synchronized gait data from a Zeno Walkway were collected and used as the ground truth to train the ERM. The ground truth gait cycle percentage was calculated from each foot’s heel strike to the next ipsilateral heel strike, as detected by the walkway. Prado de la Mora et al. detail each ERM module and accuracy of the gait cycle percentage prediction here [46].

### 2.3. Gait Phase Prediction

The predicted gait cycle percentage corresponding to the participant’s affected hemibody is input to the mTPAD force controller. We aim to apply force to the participant’s pelvis during the affected limb stance but not the contralateral stance. To determine the profile for the applied pelvic force, a sawtooth function is considered the function for the gait cycle percentage, assuming a constant gait velocity. Taking the sine of the sawtooth function, each gait cycle would encapsulate one period with a ramp-up and a ramp-down around the peak values. Using the sine-transformation of the gait cycle prediction, a smooth forcing function with a ramp-up to and ramp-down from the goal applied force magnitude can be used instead of a force step function. This is beneficial, as it can avoid sharp perturbations felt by the individual. However, the force will not be applied during the non-affected limb stance, so we only consider the positive half-cycle of the sine wave. Therefore, mapping from the gait cycle percentage to the applied pelvic force for this work will follow Equation (3) and is shown in Figure 4.
(3)$$F_{goal}=\left\{\begin{array}{cc} \left(kmg\right)*sin\left(\frac{2*\pi *p_{GC}}{100}\right), & \mathrm{if}\;p_{GC}\in \left[0,50\right)\\ 0, & \mathrm{otherwise} \end{array}\right.$$
where $F_{goal}$ is the goal force magnitude in Newtons, *k* denotes the percentage of body weight applied, *m* is the participant’s body mass in kilograms, *g* is the gravity constant, and $p_{GC}$ is the predicted gait cycle percentage from 0 to 100 of the affected side. Once the magnitude of the applied pelvic force has been calculated, we must also select the direction of this force. The mTPAD has the flexibility of using a force in any direction to the center of the pelvis. In this work, lateral and downward goal force directions are evaluated through experiments with stroke survivors, as shown in Figure 2. These force directions were selected because we hypothesized they would alter specific aspects of the ground reaction force, i.e., its vertical component as illustrated by changes in underfoot pressures and its application point as illustrated by the center of pressure.

We will investigate how the point where the GRF acts on the foot, i.e., the COP trajectory, changes when frontal plane forces are applied to the pelvis. A cyclogram is the locus of the center of pressure during a gait cycle [47]. Figure 5 shows an example of a cyclogram. It is used as a gait analysis tool. The cyclogram illustrates the spatial trajectory of the center of pressure by plotting the mediolateral (ML) displacement of the COP with respect to the direction of forward progression along the *x*-axis and the anteroposterior (AP) displacement on the *y*-axis. The COP trajectory is plotted with the two feet superimposed next to each other to visualize the symmetry of the COP throughout the gait cycle with respect to the location of each foot.

For one gait cycle that starts at the right heel strike, the center of pressure makes one complete butterfly-like cycle. The gait cycle begins with the right foot’s initial contact, which is shown in the upper left of Figure 5. As the individual shifts the weight to the right foot and off the left foot, the COP diagonally follows the gray arrow toward the bottom right. When the left foot’s toe-off occurs, the individual is in the single right support. The COP then travels forward along the foot as the individual’s GRF propels the center of mass (COM) forward. At the upper right, the left foot’s heel strike occurs. The COP then shifts toward the posterior region of the left foot, as shown on the diagonal gray arrow. When the right foot’s toe-off occurs, the individual is in a left single stance, and the COP travels forward through the left foot.

The point at which the double stance diagonal COP segments cross is the COP intersection point (CISP). This point can be used as a measure of symmetry. If the COP trajectory along one foot is shorter, the COM forward displacement is less. This asymmetry in gait would alter the CISP point, laterally shifting it toward the affected side. For instance, the top portion of Figure 5 shows a laterally symmetric gait, as the CISP is located midway between the two SS trajectories. However, for individuals with GRF asymmetry, such as stroke survivors, the CISP is shifted laterally toward the affected side, as shown in the bottom of Figure 5. By looking at different features of the cyclogram and the underfoot pressure, we can investigate how frontal plane forces applied to the pelvis affect the gait of stroke survivors.

## 3. Frontal Plane Pelvic Forces Characterization

An overground walking dataset of different applied forces on the pelvis was collected to determine the mTPAD’s ability to target stroke survivors’ overground gait asymmetry. The protocol and data collected were designed to investigate: (i) the feasibility of using the mTPAD as a robotic gait assessment and training tool for stroke survivors, (ii) the effects of a lateral force applied at the pelvis during the affected side stance, and (iii) the effects of a downwards force applied at the pelvis during the affected side stance. By understanding how these forces alter the underfoot pressure and trajectory of the COP, we can evaluate the mTPAD’s potential as a gait-training tool for stroke survivors.

### 3.1. Experimental Setup

Figure 3 illustrates a stroke survivor within the mTPAD system and the other modules of the experiment test bed, such as the instrumented walkway and the DeepSole system. The DeepSole system outputs the left and right gait cycle percentages at 40 Hz, and the mTPAD force controller optimizes for cable tensions at 40 Hz. Spatial and temporal gait parameters are calculated using a Zeno Walkway, which records gait parameters at 120 Hz. A custom sync box time-synchronized all data sources.

### 3.2. Protocol

Five stroke survivors participated in the following experiment (3M, 2F, age: 49.7± 4.8 years, height: 176.7±11.1 cm, weight: 76.1±14.7 kg). Inclusion criteria for the participants of this study were as follows: (i) above 18 years of age, (ii) diagnosis of stroke, (iii) self-reported difficulty in walking, and (iv) can safely walk with or without a cane or brace on a flat surface for 10 min at a time without pain. Before the experiment began, each participant was familiarized with the protocol and given the opportunity to ask any questions before signing a written consent form. The Institutional Review Board (IRB) of Columbia University approved the written consent form, protocol IRB-AAAT7862. Prior to starting, each participant donned a pair of comfortable shoes that housed the DeepSole system. Whole shoe sizes were available, and a snug comfortable fit was ensured. The pelvic belt of the mTPAD was placed at the level of the iliac crests, and the walker height was set to a self-selected comfortable height between 36 and 41 inches.

The protocol for this experiment, as illustrated in Figure 6, included the following conditions: (i) Baseline: overground walking with the mTPAD applying minimized force, (ii) Lateral: overground walking with the mTPAD with a lateral force applied to the pelvis, (iii) Downward: overground walking with the mTPAD with a downward force applied to the pelvis, and (iv) Post: overground walking with the mTPAD applying minimized force. After Baseline, each participant was given a short break so we could retrain the gait phase prediction model with the participant’s baseline data. The order of conditions (ii) and (iii) was randomized, and at least a five-minute seated break was given between these conditions to minimize crossover effects.

Participants walked back and forth along the Zeno Walkway under all conditions. The following laps and steps (avg ± std) were included for the training and analysis per condition: Baseline, 15.0±6.8 laps with 188.2±78.0 steps; Lateral, 27.8±7.7 laps with 354.3±83.8 steps; Downward, 27.8±10.7 laps with 340.3±101.3 steps; and Post, 14.8±5.6 laps with 191.6±66.3 steps.

After completing the Baseline, gait mat data were processed with the accompanying PKMAS software. The left and right footsteps were labeled, and the timestamps for the right heel strikes were calculated. These data and the time-synchronized raw DeepSole IMU and pressure data were added to the training dataset as illustrated in Figure 6 and retrained the prediction model for 50 epochs. Therefore, data from the participant and all prior participants were included in the training dataset, i.e., participant N’s model had the baseline data from participants 1, 2, …, and N. This strategy was used because the prediction model presented in [46] did not include data from stroke survivors walking inside the mTPAD. By including each participant’s baseline data in the model, we aimed to tailor the prediction to that individual’s gait.

### 3.3. Segmentation

Once we collected data, the gait cycle percentage and frontal plane force applications were characterized. We evaluated the changes in the COP trajectories and foot pressures. We segmented and averaged all cyclic data to obtain a representative gait cycle. We calculated the gait parameters, foot pressures, COP, and cyclogram data using the ProtoKinetic software, PKMAS [48]. Left and right heel strikes were determined as the instances each foot’s pressure became non-zero.

### 3.4. Cyclogram and CISP

The continuous COP with respect to the mat was calculated using the PKMAS mat software. The cyclogram per stride, as segmented by the affected heel strike, was calculated per stride per trial per participant. The Euclidean distance between the first and last points of the COP single stance trajectory was calculated per foot and defined as the single stance COP distance. The ML and AP positions of the CISP per stride were also computed using the PKMAS mat software. An example of these cyclogram measures is shown in Figure 5.

### 3.5. Foot Pressure

The foot pressures of the affected and non-affected limbs are studied to determine changes in weight-bearing. Total foot pressure values for the affected and non-affected limbs in relative pressure are sampled at 120 Hz, output from PKMAS, and processed using a custom Python script. The foot pressures for the affected and non-affected sides are time normalized using the affected side’s heel strikes and interpolated to 100 points. An example of the average affected and non-affected foot pressures per stride for one participant is shown in Figure 7. The peak value per stride is taken to determine if the mTPAD forces applied to the pelvis alter the maximum foot pressure. Since the pressure value is a relation between the magnitude of the force applied by the foot and the area of the foot, if a downward component of force is applied to the pelvis by the mTPAD, we would expect the pressure to increase since the area of the foot will remain the same.

### 3.6. Statistical Analysis

The Baseline data are compared to each mTPAD force condition data to evaluate the effects of timed frontal plane forces on the center of pressure trajectory and foot pressures. Before selecting each variable’s statistical test, the variable’s data distribution and normality were evaluated by a one-sample Kolmogorov–Smirnov normality test. When significantly different from a normal distribution, a Wilcoxon signed-rank test was used. Paired *t*-tests were used when not significantly different from a normal distribution. All tests were run using Python Statsmodels [49] and Scipy Stats, and statistical significance was defined as p<0.05. The following notation is used for statistical comparisons: *: p<0.05; **: p<0.01; and ***: p<0.001.

## 4. Results

The mTPAD has the unique ability to apply three-dimensional external forces on the pelvis synchronized with the user’s gait. To determine the mTPAD’s ability to target specific gait needs, we investigated these measures to determine if downward or lateral forces applied by the mTPAD encouraged increased weight-bearing during the affected limb stance as hypothesized.

### 4.1. Frontal Plane Force Application

Applied forces to the pelvis were regulated based on a BW% and thresholding scheme. A maximum tolerable force BW% was determined per participant for each force condition. Force application started at 2% body weight and incremented by 2% until either 10% was reached or the participant could not tolerate the force, as verbally expressed. The group forces (avg ± std) are 7.2%±2.3% during the Lateral condition and 8.8%±1.8% for the Downward condition. Each participant’s body weight and percentage per direction are shown in Table 1.

### 4.2. Cyclogram and CISP

The spatial characteristics of the cyclogram are shown in Table 2. Step width and SS COP distances for both the affected and non-affected sides are reported per participant and trial. The ML CISP reflects the lateral symmetry of the COP trajectory, and the group results and results per participant are shown in Figure 8. ML CISP values were normally distributed for the Baseline and both force conditions, so paired *t*-tests were used to investigate the changes from the Baseline. ML CISP % is shifted significantly closer to the unaffected side when a lateral force is applied to the pelvis by the mTPAD (5.5%±53.7%) than when no forces are applied by the mTPAD (−9.8%±54.1%); t(4) = 3.0215, p=0.0391, a total shift of 15.3%. When a downward force is applied (0.8%±52.6%), there is no significant change from the baseline; t(4) = 1.4740, p=0.2145. These results match the hypothesis that a lateral force would alter the lateral shift of the COP trajectory.

### 4.3. Foot Pressures

To determine changes in weight-bearing during applied pelvic forces, the maximum pressure for both the affected and non-affected limbs is evaluated. The group results and results per participant for maximum relative pressures are shown in Figure 9 and are relative pressure values output from the instrumented walkway’s pressure sensors. Both sides’ values were normally distributed for the Baseline and both force conditions, so paired *t*-tests were used to investigate the changes from the Baseline due to each force condition. When no external force is applied to the pelvis by the mTPAD, the mean maximum pressure value per stride is (233.7±61.5p) for the affected side and (241.9±47.2p) for the non-affected side. When a lateral force is applied by the mTPAD to the pelvis, the maximum pressure on the affected side does not change (225.6±58.4p); t(4) = 2.0899, p=0.1048, and neither does the maximum pressure on the contralateral side (237.8±47.8p); t(4) = 2.5049, p=0.0664. When a downward force is applied, the maximum pressure on the affected side increases compared to baseline (234.1±59.2p); t(4) = 4.5375, p=0.0105, but there is no significant difference on the non-affected side (244.9±44.5p); t(4) = 1.9600, p=0.1216. While the change in the grouped results may appear slight for the Downward condition, the pairwise changes illustrate that all participants had an increase in mean maximum pressure on the affected foot. These results are expected, as a downward force during the affected stance should increase the magnitude of the vertical ground reaction force, increasing the weight bearing during the affected limb stance.

## 5. Discussion

Stroke survivors, or individuals with any gait pathology that has altered ground reaction forces, benefit from gait therapy that aims to improve weight bearing and symmetry through the gait cycle. By encouraging weight bearing during overground gait, we may alter the GRF by changing its magnitude or placement under the feet. Improving gait quality and even independence in daily life activities may be possible through interventions that induce these changes. Stroke survivors and others may benefit from a training paradigm targeting specific gait deficits during overground walking, which is a unique ability demonstrated by the mTPAD.

Here, the mTPAD was used to apply a frontal plane pelvic force during the affected side stance for a group of stroke survivors. The ability of the mTPAD to apply custom forces timed with the user’s gait cycle percentage gives the mTPAD the unique ability to tailor these frontal plane forces to the user’s gait, even when the user’s gait is asymmetric.

When a force is applied laterally to the pelvis during the affected stance, the lateral symmetry of the COP trajectory changes. A maximum of 10% BW was used, with 7%±2% BW force achieved by the group of stroke survivors walking overground. This force significantly shifted the COP intersection point toward the symmetric center, indicating a lateral shift of the COM during the affected limb stance. This illustrates that when the mTPAD applies a lateral force to the user’s pelvis, a more laterally symmetric COP trajectory is achieved. The lateral force applied by the mTPAD does not significantly alter the foot pressures for the unaffected and affected side of the group of stroke survivors.

When a downward force is applied to the pelvis during the affected stance, the underfoot pressure maximum increases for the affected side. In this force condition, 9%±2% BW force was applied downward at the pelvis, which is a higher average force than the lateral condition. This increase was not seen bilaterally, as the foot pressure of the unaffected limb did not change. Therefore, by applying a downward force during the affected stance, the mTPAD can increase weight bearing during the affected stance. The lateral symmetry of the COP does not change as it did during a lateral force application.

To better understand these changes to the characteristics of the GRF when two forces are applied at the pelvis, let us consider a simple quasi-static inverted pendulum model, as shown in Figure 10. When considering the simplified inverted pendulum model of a single stance with the mass at the end of the leg length, we can calculate the GRF and medial–lateral location of the COP. When no external force is applied to the pelvis, the vertical GRF magnitude equals the body weight, and the COP is located vertically beneath the COM. When a lateral force is applied at the COM, the COP shifts laterally, and this displacement is related to the magnitude of the lateral force. Considering a 10% BW force applied laterally at the pelvis, the theoretical change to the COP is $0.1d_{z}$ away from the medial center. When a force is applied downward to the COM during a single stance, the lateral displacement of the COP does not change. However, the magnitude of the GRF does increase to match the relative increase of the resultant downward force.

Therefore, the changes to the underfoot pressure and ML CISP made by the mTPAD’s applied forces can be motivated by the theoretical modifications to the GRF components. During an applied lateral force, the lateral shift of the COP is reflected in the significantly increased symmetry of the COP trajectory without significantly changing the maximum pressure value. For the downward force, the maximum pressure significantly increases for the affected side, with no significant change to the unaffected side’s maximum pressure or lateral symmetry of the COP trajectory. Thus, the forces applied to the pelvis at the mTPAD can alter the GRF’s characteristics, making it a promising tool for overground gait training. This robotic platform allows us to motivate different force applications to target specific loading responses to tailor the prescribed overground force intervention to each individual’s gait deficits and needs.

While this work highlights the mTPAD’s ability to alter specific components of the GRF during a group of stroke survivors’ overground gait, this work is not without its limitations. The interaction between the mTPAD and the participant’s arms was not scientifically measured and could act as a distribution path for the applied force to the ground [50]. However, forearm armrests may limit participants from leaning on the mTPAD, mitigating loading through the device. These armrests, visible in Figure 3, also ensured comfortable arm placements and hand-grasping of the mTPAD, especially for those with spasticity in the affected arm. It is possible that the variability and asymmetry of the stroke suvivors’ gaits altered the accuracy of the gait cycle prediction model, which has been shown to have an RSME of 7.2% in neurotypical individuals. We included each participant’s baseline gait data in the trained model to mitigate this risk. With the tailored model, the theoretical changes based on the inverted pendulum model were seen in the experimental data. The changes made during this single-session study are promising, but further work must be conducted to determine the last effects of these gait-synchronized forces on stroke survivors and their independence. Other caveats of this study include the limited number of stroke survivors participating in the experiment and a lack of measured lower limb kinematics. While the mTPAD does not measure lower limb kinematics, it also can be used independently from a motion capture lab setting, making it a portable device that can be used anywhere. It is also possible that in future versions of this work, the DeepSole system can be used to provide feedback regarding the change in underfoot pressure in real time. Therefore, this work still highlights the ability of the mTPAD to alter the vertical magnitude and application point of the GRF, which is a critical goal for post-stroke individuals.

## 6. Conclusions

This work demonstrates the efficacy of using the mTPAD to alter GRF characteristics of stroke survivors—a critical part of increasing the gait ability and quality of life for these individuals. We applied two frontal plane forces that encouraged weight bearing and lateral symmetry during overground affected limb stances. We evaluated these forces experimentally with a group of stroke survivors. The lateral force reduced the ML CISP, increasing overground gait symmetry. The downward force increased the maximum relative pressure under the affected limb. These applied lateral and downward forces show promising benefits that could improve the ability to bear weight on the affected limb through a multi-session training regimen. This work opens the door to overground robotic gait training protocols to enhance lateral symmetry and weight bearing for individuals with these deficits. The potential of the mTPAD as a gait-training tool can now be evaluated and even compared against traditional gait therapy.

Stroke survivors, or individuals with any gait pathology that has altered ground reaction forces, benefit from gait therapy that aims to improve weight bearing and symmetry through the gait cycle. By altering the GRF, either by changing its magnitude or placement under the feet, it may be possible to improve gait quality and even independence in daily life activities. In addition, these individuals may benefit from a training paradigm that can target the COPs during overground walking, which is a unique and proven ability of the mTPAD.

## Figures and Tables

**Figure 1 bioengineering-10-00698-f001:**
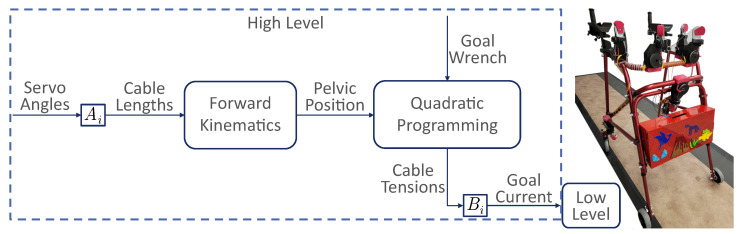
Illustrative diagram for the mTPAD’s open-loop controller and the mTPAD system. $A_{i}$ maps the servo angle to the cable length for each motor. The pelvis position relative to the mTPAD frame is determined using forward kinematics [44]. The pelvis position and the goal wrench are input to the quadratic programmer used to optimize cable tensions. $B_{i}$ maps the cable tensions to the servo currents. A low-level PID controller is implemented on the servo currents at the motor level.

**Figure 2 bioengineering-10-00698-f002:**
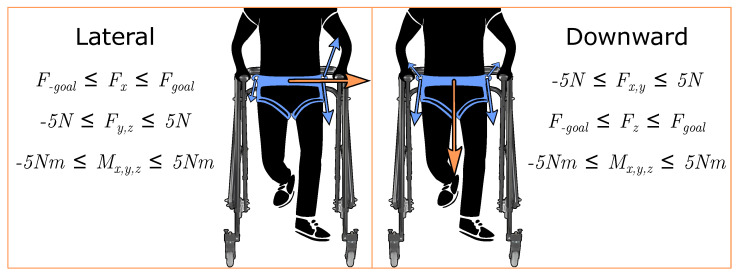
Forces applied to the pelvis and their associated quadratic programming wrench constraints. Here, *x* refers to the mediolateral axis and *z* refers to the vertical axis. For the lateral force, $F_{x}$ is limited within a small range of $F_{goal}\pm 0.5$ N. All other forces and moments are limited within ±5 N or Nm. These ranges were selected to minimize forces in other directions while ensuring that the goal lateral and downward forces could be achieved by the mTPAD’s controller. Similarly for the downward force, $F_{z}$ is limited within ±0.5 N from $F_{goal}$. The orange arrows depict the frontal plane pelvic forces that are applied by the mTPAD. The blue arrows depict the cable tensions, with the relative size representing the relative tension required to apply the pelvic forces. These pelvic forces are synchronized with the user’s gait cycle percentage corresponding to single stance. All seven cables are actuated to minimize all other forces and moments.

**Figure 3 bioengineering-10-00698-f003:**
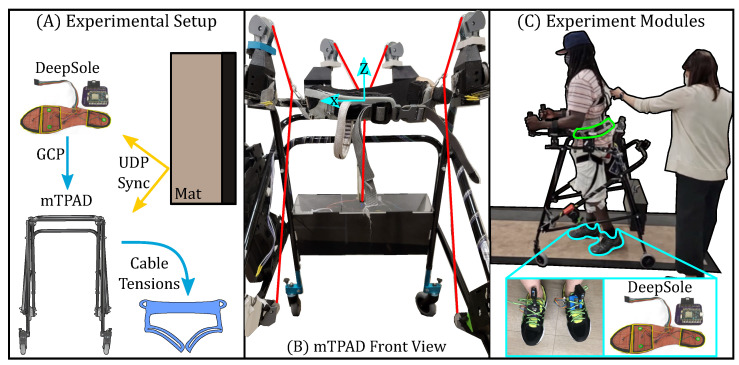
Experimental setup for the single session experiment evaluating effects of external forces applied to the pelvis on the gait of stroke survivors. (**A**) The DeepSole system sent the predicted gait cycle percentage to the mTPAD in real time. The predicted GCP of the affected side was an input to the mTPAD’s open-loop controller, which outputs each cable’s optimized tension. These data were time synchronized with the instrumented mat via UDP. (**B**) The front view of the mTPAD illustrates the local coordinate frame used and the seven cables that route from the frame to the pelvic belt. (**C**) A participant is walking overground in the mTPAD. Each participant walked on a Zeno Walkway. The DeepSole system is outlined in blue and the mTPAD pelvic belt is highlighted in green.

**Figure 4 bioengineering-10-00698-f004:**
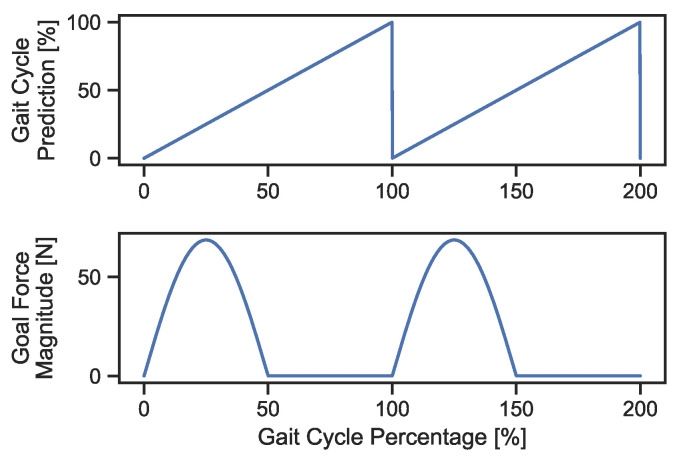
The top graph illustrates the theoretical gait cycle percentage as a sawtooth function for two full gait cycles. The lower graph shows the corresponding goal force magnitude calculated by Equation (3) for a 70-kg adult. At the affected side heel strikes, shown here as 0% and 100%, the force magnitude increases to a peak of 70 Newtons at 25% and 125%. After these points, the magnitude decreases and reaches 0 Newtons at 50% and 150% of the gait cycle.

**Figure 5 bioengineering-10-00698-f005:**
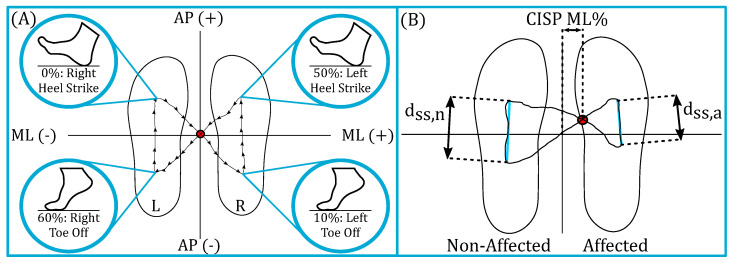
COP Cyclogram and Cyclogram Characteristics. (**A**) A symmetric COP cyclogram with gait event labels. The *x*-axis of the cyclogram is the ML direction, and the *y*-axis is the AP direction. The SS (vertical) COP trajectories along the left and right feet are approximately the same length. The CISP, represented by a red circle, lies approximately at (0,0) on the cyclogram. This means the individual can comparably progress their center of mass forward during their left and right stances. (**B**) An asymmetric COP cyclogram with labeled characteristics. The right foot’s COP traveled distance is shorter than the left foot’s, forcing the ML CISP laterally toward the affected side. This illustrates that the individual’s ability to propel themselves forward during their single right stance is impaired compared to the left side. Therefore, the lateral placement of the CISP on the COP cyclogram represents the symmetry of generated single-stance GRFs and the overall COP trajectory.

**Figure 6 bioengineering-10-00698-f006:**
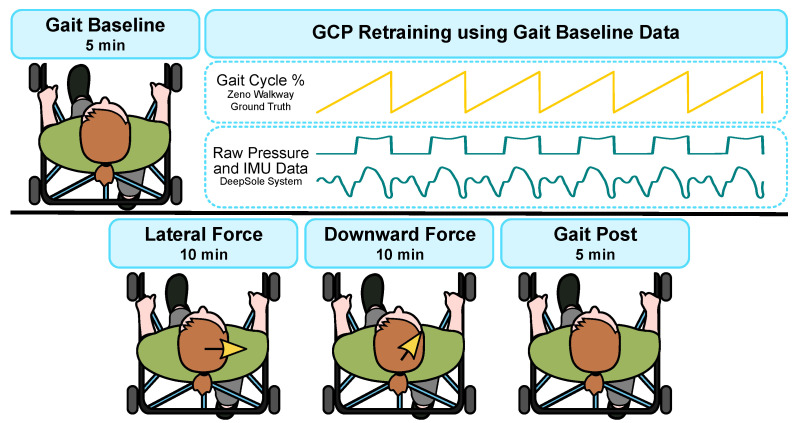
An illustration of the protocol used in this work. The protocol started with a gait Baseline. The DeepSole and Mat data were then used to retrain the ERM model. This new prediction model was used for the last three sessions. By retraining the model with each participant’s gait Baseline, the prediction is tailored to their gait and has training data with irregular or asymmetric gait cycles.

**Figure 7 bioengineering-10-00698-f007:**
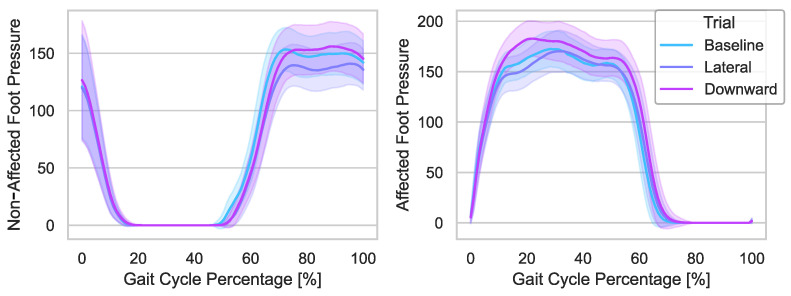
The *x*-axis is the gait cycle percentage segmented by right heel strikes. The *y*-axis is the pressure of the affected (right foot, bottom plot) and non-affected (left foot, top plot) feet. The solid lines represent each condition’s average pressure curve for all strides for this participant, and the shaded regions around each line represent the standard deviations.

**Figure 8 bioengineering-10-00698-f008:**
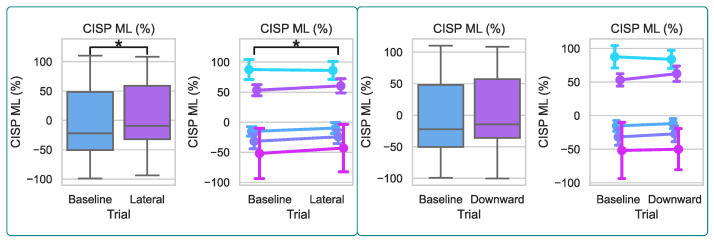
ML CISP Percentage per Condition. The left plots represent the lateral force, and the right plots show the downward force. The left plot in each set shows the group differences. The right plot in each set shows the mean and standard deviation per participant, where each color denotes a different participant. The following notation is used for statistical comparisons: *: p<0.05; **: p<0.01; and ***: p<0.001.

**Figure 9 bioengineering-10-00698-f009:**
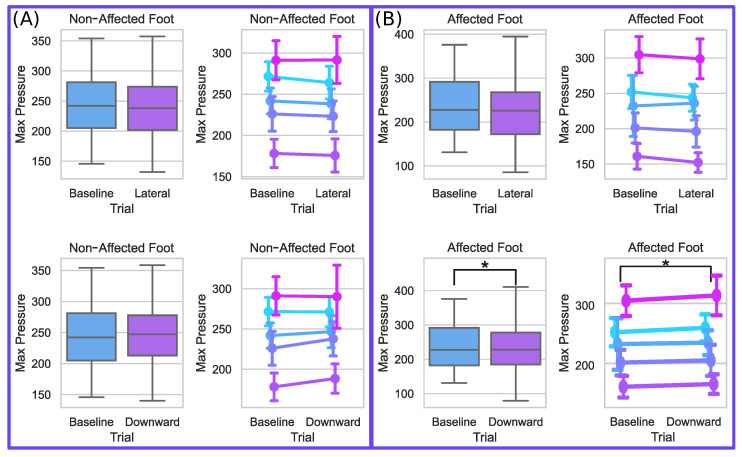
Non-Affected and Affected Max Pressure per Condition. (**A**) Non-affected side foot pressure per force condition. The top row represents the lateral force, and the bottom row shows the downward force. The left column shows the group differences. The right column shows the mean and standard deviation per participant. (**B**) Affected side foot pressure per force condition. The top row represents the lateral force, and the bottom row shows the downward force. The left column shows the group differences. The right column shows the mean and standard deviation per participant, where each color denotes a different participant. The following notation is used for statistical comparisons: *: p<0.05; **: p<0.01; and ***: p<0.001.

**Figure 10 bioengineering-10-00698-f010:**
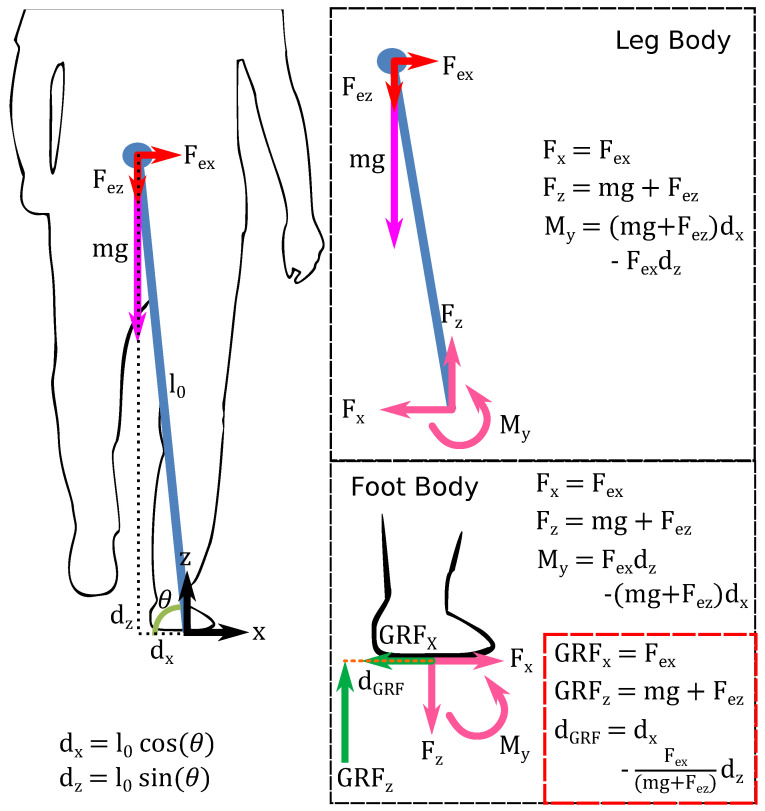
Free Body Diagram with External Applied Forces to the Pelvis. Basic free body diagram for the center of mass while in single stance, as an inverted pendulum. The first segment, i.e., the leg body, shows the external and internal forces on the leg, balancing the gravitational force applied at the center of mass. The second segment, i.e., the foot body, shows the equal but opposite forces and moments at the ankle and the location and magnitude of the ground reaction force created. This model illustrates that if $F_{ex}=0$, the COP is the projection of the COM onto the horizontal plane.

**Table 1 bioengineering-10-00698-t001:** Body weight % and force applied per force direction.

Participant	1	2	3	4	5
**BodyWeight (kg)**	79.4	76.2	68.0	52.2	94.3
**Lateral (BW%)**	4%	10%	6%	8%	8%
**Lateral Force (N)**	31.1	74.7	40.0	40.9	74.0
**Downward (BW%)**	6%	10%	10%	10%	8%
**Downward Force (N)**	46.7	74.7	66.7	51.2	74.0

**Table 2 bioengineering-10-00698-t002:** Mean Cyclogram Characteristics per Participant: Step Width (SW), Affected (Af) SS COP Distance, Non-Affected (Naf) SS COP Distance.

Participant	1	2	3	4	5
**Baseline (B) SW (cm)**	5.1	8.6	3.6	13.2	13.3
**Lateral (L) SW (cm)**	6.1	9.4	4.8	15.4	12.7
**Downward (D) SW (cm)**	4.9	9.0	5.8	14.2	11.9
**B Af SS COP Dist. (cm)**	1.8	7.4	6.4	5.2	1.8
**L Af SS COP Dist. (cm)**	1.7	8.3	7.1	4.5	1.8
**D Af SS COP Dist. (cm)**	1.8	8.6	6.7	4.1	2.6
**B Naf SS COP Dist. (cm)**	12.2	11.0	14.9	13.7	3.3
**L Naf SS COP Dist. (cm)**	11.8	10.9	14.1	13.1	4.1
**D Naf SS COP Dist. (cm)**	12.5	11.4	14.4	12.7	3.8

## Data Availability

The data presented in this study are available in this article.

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
