# Peer review of "Overground Robotic Gait Trainer mTPAD Improves Gait Symmetry and Weight Bearing in Stroke Survivors"

_bioengineering, 2023, doi:10.3390/bioengineering10060698_

Round 1

Reviewer 1 Report

The authors presented the gait symmetry and foot pressure during walking after applying lateral and downward forces using a robotic gait trainer mTPAD. Gait asymmetry is a problem for individuals with stroke, and the authors were trying to address this challenging issue.

Overall, The paper was well developed with detailed descriptions of the robot, the experimental design, and the results.

Some minor suggestions:

1) line 65:“Although the literature on post-stroke robotic gait training is mixed.” It would be better to elaborate on the mixed findings.

2) The study’s objective can be described more explicitly in the background session;

3) Figure 2. It is unclear why 5N and 5Nm were set as the forces and moments limit.

4) It is unclear what statistical method was used for the results. It seemed like t-tests were used. However, given the small sample size and large variations, I suggest using non-parametric tests.

5) line 326-328: while the differences in maximum pressure force of the affected side were statistically significant, the changes were small. It would be better to discuss the clinical meaning of the differences.

Reviewer 2 Report

In this interesting work, the Authors used a previously developed robotic device for the application of pelvic forces on stroke survivors, in order to improve their gait stability and symmetry by altering their GRF characteristics, through a forward kinematic control. Gait characteristics are assessed using instrumented shoes and a deep learning model for gait cycle percentage prediction.

The Authors found an improvement of the mediolateral COP intersection point, corresponding to an increase in gait symmetry, by applying lateral forces, while downward forces increased maximum relative pressure under the affected limb. The work is well structured and the methods and results are clearly presented. However, I have two questions:

1. when comparing to baseline measurements, did the Authors take into account any p value correction (e.g., fdr)?

2. how about implementing a closed control loop, by feeding information from instrumented shoes?

English is fine, just a few mistypes (e.g. in Fig.2 caption)
